# The Role of Mutant p63 in Female Fertility

**DOI:** 10.3390/ijms22168968

**Published:** 2021-08-20

**Authors:** Yi Luan, Pauline Xu, Seok-Yeong Yu, So-Youn Kim

**Affiliations:** Department of Obstetrics and Gynecology, Olson Center for Women’s Health, College of Medicine, University of Nebraska Medical Center, Omaha, NE 68198, USA; yi.luan@unmc.edu (Y.L.); pauline.xu@unmc.edu (P.X.); syu@unmc.edu (S.-Y.Y.)

**Keywords:** p63, oocyte, mutant, infertility

## Abstract

The transcription factor p63, one of the p53 family members, plays an essential role in regulating maternal reproduction and genomic integrity as well as epidermal development. *TP63 (human)*/*Trp63 (mouse)* produces multiple isoforms: TAp63 and ΔNp63, which possess a different N-terminus depending on two different promoters, and p63a, p63b, p63g, p63δ, and p63ε as products of alternative splicing at the C-terminus. TAp63 expression turns on in the nuclei of primordial germ cells in females and is maintained mainly in the oocyte nuclei of immature follicles. It has been established that TAp63 is the genomic guardian in oocytes of the female ovaries and plays a central role in determining the oocyte fate upon oocyte damage. Lately, there is increasing evidence that *TP63* mutations are connected with female infertility, including isolated premature ovarian insufficiency (POI) and syndromic POI. Here, we review the biological functions of p63 in females and discuss the consequences of p63 mutations, which result in infertility in human patients.

## 1. Introduction

The p53 family proteins play essential roles in maintaining maternal reproduction and genomic integrity, and these essential roles appear to be conserved across different organisms from invertebrates to mammals. A p63/p73 hybrid gene exists in the genome of unicellular flagellates and functionally acts to preserve the genome stability of germ cells in sea anemones [1]. In addition, recent reports of increased rates of embryonic implantation failure and miscarriage in women with p53 gene polymorphism indicate the importance of p53 in female fertility and in vitro fertilization (IVF) success, although the potential molecular mechanisms remain to be elucidated [2]. 

While the separation between p63 and p73 is observed in vertebrates, these proteins retain highly conserved female germline-protecting roles with distinct mechanisms in oocytes [3]. TAp63 is highly expressed in the oocytes of primordial follicles, which represent the ovarian reserve and support female fertility with little to no significant expression in preovulatory follicles [4]. When DNA damage occurs, DNA double-strand breaks (DSBs) lead to the activation of TAp63, and oocyte apoptosis ensues [5]. 

TAp73 was shown to mediate chromosome alignment during oocyte maturation in the ovaries, control ovulation, and manage embryonic development during preimplantation [6]. Aging-related maternal infertility has been associated with defects in chromosome alignment and embryonic development [7]. Therefore, these observations suggest a possible implication of TAp73/ΔNp73 in aging-related infertility in women.

The ovarian reserve is the pool of primordial follicles, which determines the reproductive lifespan of females. A diminished ovarian reserve (DOR) in women indicates a loss of their reproductive potential and is one of the leading causes of infertility [8]. TAp63 is a key regulator of the genome stability of the female germline to achieve faithful transmission of genetic information to the next generation. Conversely, mutations and/or inactivation of TAp63 can induce poor quality of the ovarian reserve, leading to infertility. Therefore, this review will focus on the biological roles of TAp63 in oocytes and its association with fertility.

## 2. Structure of the p53 Family Members

The transcriptional p53 family members consist of p53, p63, and p73, which have a common structural backbone composed of transactivation (TA), DNA-binding domain (DBD) and oligomerization domain (OD) with differences at the C-terminal region. Unlike p53, which has an unstructured C-terminus, p63 and p73 contain additional domains at the C-terminus. p63 contains a secondary TA domain, sterile alpha motif (SAM) domain, and transactivation inhibitory domain (TID), whereas p73 contains only a SAM domain and TID (Figure 1A). 

These three proteins are present as inactive forms before adopting an active conformation following multiple phosphorylation events. When p53 and p73 are inactive, they present as tetrameric structures without any phosphorylation. However, p63 forms a tetramer from two dimers following multiple phosphorylations by kinases [9] (Figure 1B). These proteins (p53, p63, and p73) possess an alternative promoter at the N-terminus, generating N-terminally truncated isoforms (Δ133Np53, ΔNp63, and ΔNp73) as well as isoforms where the N-terminal TA domain is omitted from the complete structures (p53, TAp63, and TAp73) [10] (Figure 1C). 

The absence of the N-terminal TA domain is, thus, expected to not only generate N-terminally truncated isoforms inert as a transcriptional factor but also to act as antagonists toward their corresponding isoforms. In addition, alternative splicing at the C-terminus during transcription yields several variants [10,11]. For example, TAp63 itself has been known to possess five variants, α, β, γ, δ, and ε [11] (Figure 1C). TAp63α represents the predominant form of TAp63 in oocytes and is transcribed from exon 2 to exon 14, containing all the domains at the C-terminus [12]. 

TAp63β is generated with exon 13 skipped, and TAp63γ lacks exon 11 through exon 14, which encode the SAM domain and TID. TAp63δ skips exons 12 and 13, while TAp63ε is derived from exons 2 through 10 with a 5’ portion of intron 10 transcribed. Structurally, the three main domains of TAp63 (DBD, the TA domain, and the SAM domain) remain folded in a quiescent state. Deletion mutagenesis further suggests that the last 70 amino acids of TAp63α constitute a transactivation inhibitory domain (TID) against the N-terminal TA domain, thereby, forming a closed and inactive conformation [9].

The TID contains two distinct inhibitory subdomains with independent regulatory functions [13]. One subdomain is based on sumoylation, which affects the intracellular protein concentration. The other subdomain is a stretch of ~13 highly conserved amino acids shown to be responsible for auto-inhibiting intramolecular interaction. Further analyses [14] reported that C-terminal Ser-Gln and Thr-Gln (S/TQ) sites are targets of phosphorylation for protein activation. In addition, TAp63α phosphorylation at the C-terminal S/TQ sites appeared to require the N-terminal TA domain. 

Therefore, the generation of multiple variants at the transcriptional level would influence the transcriptional potentials of p53 family proteins and suggests that the relative ratio among isoforms would reflect the overall activity of each p53 family protein at the cellular level when they are co-expressed locally.

## 3. p63 Expression and Regulation in the Ovarian Germ Cells

Our current understanding of the biological roles of TAp63, commonly referred to as the guardian of the female germline, in the genomic integrity of female germ cells has been mostly obtained using mouse models [4,15]. In mice, TAp63 expression has been detected in diploid mitotic primordial germ cells (PGCs) as early as embryonic day 7.5 (E7.5), after which PGCs migrate to the genital ridge at E10.5 via the hindgut [16]. Afterward, the number of PGCs increases and p63 expression becomes prominent in the genital ridge, which will later develop into gonads. 

When oogonia undergo meiosis between E13.5 and E16.5, they possess multiple double-strand breaks (DSBs) due to recombination between chromosomes. The expression of p63 becomes suppressed in this time frame, supporting the observation that PGCs in germ cell clusters withstand hundreds of programmed DNA DSBs required for meiotic recombination due to the lack of TAp63 expression. However, 4–10 DSBs induced by γ-radiation treatment or other routes of DNA damage are sufficient to induce apoptosis in postnatal immature oocytes due to the expression of TAp63 [17]. 

The number of TAp63-positive oocytes and the intensity of TAp63 expression begin to increase from E17.5 to postnatal day 5 (PD5) when most of the oocytes have entered meiotic arrest (Figure 2). After birth, the expression levels of TAp63 remain high in the oocytes of primordial, primary, and early secondary follicles, and these oocytes are α form-positive while the expression of its Δ isoform remains undetected from E10.5 to adulthood [4,16,18]. 

This distinct expression pattern of TAp63 in female germ cells during embryonic development and after birth raised questions about whether TAp63 expression is essential for the development of female germ cells. Several groups have demonstrated that ovaries from p63 null embryos developed similarly to wild-type ovaries and exhibited a histologically normal phenotype with primordial follicles, growing follicles that appear following recruitment for ovulation, and corpora lutea [4]. 

This supports that the notion that TAp63 is dispensable for the development of oocytes, follicles, and ovulation although it is expressed during the narrow time window of germ cell development. Interestingly, PGCs in the ovaries of the human fetus at post-last-menses (PLM) 12 weeks did not express p63; however, p63-positive PGCs gradually increase between PLM12 and 16 weeks [4].

All PGCs reach dictyate arrest by postnatal day 5 (PD5) [19]. It has been reported that, while TAp63α expression was relatively uniform in PD5 oocytes, TAp63α-lacking predictyate oogonia in the newborn PD0.5 ovaries contained more γ-H2AX foci than primary oocytes at the dictyate stage showing TAp63α expression. Thus, meiotic DNA DSBs and TAp63α expression appear, at some level, to be inversely correlated in newborn primary oocytes [17]. 

Since predominantly PD5 primary oocytes express only TAp63α and lack γ-H2AX foci, either all meiotic DNA DSBs undergo repair by PD5 dictyate arrest or those that fail to undergo repair undergo TAp63α-mediated death [20]. Whereas PD5 oocytes of the immature primordial follicles are totally obliterated within only 2 days of ionizing radiation (IR) treatment, a significant number of PD0.5 immature oogonia possess the capacity to survive IR-induced DNA damage for at least 7 days after IR treatment due to the absence of TAp63 [5]. 

While irradiated PD5 oocytes demonstrated a prominent TAp63 phosphorylation shift, irradiated PD0.5 oocytes did not. However, IR-induced γ-H2AX levels increased following IR treatment in both PD0.5 and PD5 ovary lysates, indicating that DNA damage was incurred in both PD0.5 and PD5 ovaries [17]. Thus, oogonia appear to quickly acquire the potential to induce TAp63α phosphorylation after birth [15].

It has been demonstrated that oocytes in the primordial follicles of mice lacking exons 2 and 3 of the *Trp63* gene encoding the transactivation domain-specific N-terminus were resistant to equivalent doses of IR treatment that killed virtually all oocytes of primordial follicles in wild-type ovaries [12]. Similar results were observed when comparing wild-type ovaries grown and irradiated in vitro with ovaries isolated from TAp63-null mice. 

The association observed between TAp63α phosphorylation and oocyte cell death was the first to suggest that TAp63α was phosphorylated in response to DNA damage [15]. Phosphorylated TAp63α displayed an increase in DNA binding to nearly 20-fold higher than that of non-irradiated oocytes. Further studies confirmed that IR treatment rapidly triggered TAp63α phosphorylation in oocytes. The p63 null mutation, which did not affect the spontaneous development of oocytes and ovarian follicles, protected mouse oocytes from IR-treatment-induced apoptosis by preventing the cleavage of caspases-9 and -3 as well as follicle loss induced by IR treatment [21].

## 4. Regulation of p63 Structure in the Oocyte

Size exclusion chromatography analysis of purified TAp63α demonstrated that it adopts a dimeric, inactive conformation, suggesting that the activation of p63 may be linked to the formation of tetramers. Analysis of the oligomeric state of TAp63α in ovaries of nonirradiated mice revealed a strong signal in the dimer fraction, with no detectable signal in the tetramer fraction. In contrast, ovaries obtained from irradiated mice revealed a significant signal in the tetramer fraction. Accordingly, TAp63α in non-stressed oocytes is maintained in a dimeric and closed conformation, while DNA damage triggers the formation of tetramers [9]. 

The significantly higher concentration of TAp63α in nonirradiated versus irradiated ovaries further suggested that the formation of tetramers is not suppressed by keeping the intracellular protein concentration low but actively by domain–domain interactions involving the transactivation inhibitory domain (TID). The SAM domain and DBD were described to not be essential for retaining the dimeric state. 

Later analyses with secondary structure prediction programs predicted the existence of an α-helix (TA1) and two β-strands (TA2A and TA2B) in the TA domain and a β-strand in the TID. TA2B and TID form an anti-parallel β-sheet with a polar and hydrophobic face, proving that a β-sheet covers the tetramerization interface of the tetramerization domain (TD), thus inhibiting the formation of tetramers [12]. In addition, the TA1 helix was confirmed to bind to the TD, further stabilizing the closed and compact formation. Small- angle X-ray scattering measured a dimeric structure of the TAp63α with the DBDs positioned at the outside, while the complex formed by the TA domain, TD, and TID forms the center of the molecule.

## 5. Regulation of p63 in the Oocyte

TAp63 is the central integrator for DNA damage signaling in oocytes of primordial follicles [12,20,21,22,23]. When DNA damage is detected, in particular DNA DSBs, this triggers the activation of a kinase cascade starting with Ataxia Telangiectasia Mutated (ATM) and/or Ataxia Telangiectasia and Rad3 related (ATR) to phosphorylate histone H2AX to mark the position of the lesion and activate checkpoint kinase 1/2 (CHK1/2) [5,24,25,26] (Figure 3). Later studies demonstrated that TAp63α is a direct target of activated CHK2 and that it phosphorylates S582 [20]. 

TAp63α in its active tetrameric conformation acts as a transcriptional activator, resulting in the induction of *Puma* and *Noxa* and the subsequent binding of the potent apoptosis inducers PUMA and NOXA directly or indirectly to BAX and BAK as established by Kerr et al. [27,28,29]. Recent studies suggest that checkpoint kinase 1 (CHK1) also becomes activated by persistent DSBs in oocytes and, to an increased degree when CHK2 is absent, signaling to p63 for oocyte elimination [30]. Additional studies reported that casein kinase 2β (CK2β)-deficient oocytes showed enhanced γ-H2AX signals indicative of accumulated unrepaired DSBs, which activated CHK2-dependent p63 signaling [31].

Treatment of mouse ovaries with the CHK2 inhibitor II BML-277 to prevent phosphorylation at S582 and subsequent irradiation revealed the nearly complete suppression of TAp63α and almost no formation of tetramers, thus, confirming the previously reported role of Chk2 in the activation process [32]. Additional studies also observed that chemical inhibition of ATM kinase activity blocked TAp63α phosphorylation following irradiation and preventing oocyte death [17]. Phosphorylation by CHK2 at S582 alone was determined to be insufficient for TAp63α tetramerization and, thus, activation required further phosphorylation events by additional kinases. 

Previous analyses of antibody-purified TAp63 showed clusters of phosphorylation sites in the linker regions between the TA domain and the DBD, between the TID and the SAM domain, and immediately following the CHK2 site, N-terminally to the TID [33]. In addition to S582 and S583, only amino acid residues 585–594 were demonstrated through comprehensive mutational analysis to be essential for tetramerization. This sequence stretch follows a highly conserved casein kinase 1 (CK1) consensus pattern with phosphorylatable residues located in every third position C-terminally to S582. 

All phosphorylation events up to S591 and S592 were deemed essential for efficient tetramerization with the charge density in the region directly N-terminal to the TID as the decisive factor. This stretch of phosphorylated amino acids required for the activation process is located in an unstructured loop directly preceding the TID. Located adjacently is a loop following the TAD harboring three negatively charged aspartate residues that function as the sensor domain for creating the electrostatic repulsive force leading to disruption of the previously established inhibitory β-sheet. 

Thus, CHK2 acts as a “priming kinase” at S582 and CK1 as an “executioner kinase”, which adds phosphate groups at four consecutive sites (S585, S588, S591, and T594) (Figure 3). Phosphorylation of S591 is the “point of no return” [32]. Treatment with CHK2 or CK1 inhibitor in cultured mouse ovaries directly interfering with TAp63α activation following DNA damage by doxorubicin or cisplatin rescued the majority of oocytes in primordial follicles.

The tetramerization kinetics of TAp63α following irradiation was revealed to have a sigmoidal time response, where CK1 phosphorylates TAp63α in a sequential manner and a biphasic kinetic. The first two phosphorylation events are fast, while the third one is the slowest and essential for the formation of the open and active conformation. The decrease in phosphorylation kinetics with higher phosphorylation levels may be due to a decrease in the binding affinity of the increasingly phosphorylated peptide, rendering binding in the substrate-bound position as needed for phosphorylation of S591 as less favorable [34].

## 6. p63 and Female Infertility

Infertility, also referred to as subfertility, is defined as “a disease characterized by the failure to establish a clinical pregnancy after 12 months of regular, unprotected sexual intercourse or due to an impairment of a person’s capacity to reproduce either as an individual or with his/her partner” by the World Health Organization (WHO) International Committee for Monitoring Assisted Reproductive Technology (ICMART) and fertility societies [35,36]. Common causes of infertility include ovulatory dysfunction (polycystic ovarian syndrome (PCOS) [37], premature ovarian insufficiency (POI) [38], primary ovarian failure (POF)), ovarian cysts [39], fallopian tube blockage or absence, endometriosis [40], uterine fibrosis, and poor oocyte quality [41].

The genomic basis of female infertility is rather complex and is determined by many factors. Using karyotype analysis, several chromosome aberrations have been identified to be related to female fertility. Recent studies have found additional candidate monogenetic mutations capable of causing female infertility, including *FMR1*, *BMP15*, *PGRMC1*, and *ZP3* [42]. 

Although *TP63* was not traditionally considered as a primary predictor in female infertility, later research demonstrated its involvement in regulating oocyte fate and fertility. An increasing number of reports have confirmed the role of p63 in the reproduction of mice and humans. Studies with mouse models revealed the expression of TAp63 isoforms in all female reproductive organs, with the highest concentrations in the vagina and ovaries [4,16]. *TP63*-related cases were reported recently with infertility as a component of the syndromic phenotypes or isolated premature ovarian insufficiency (POI).

### 6.1. Mouse

The roles of TAp63 and ∆Np63α in vivo have been previously reported by generating conditional knockout mouse models for either isoform, leading to distinct phenotypes. It has been demonstrated that *Trp63* is integral for murine embryo cranio-facial, skin, and limb development [43]. A ∆Np63 knockout mouse model displayed striking developmental abnormalities, which confirmed the role of ∆Np63 in epithelial biology [44]. In a TAp63 knockout mouse model, abnormal development in cellular senescence was detected, preventing premature tissue aging and precipitating defects in the lipid and glucose metabolism [45,46]. 

Both ∆Np63 and TAp63 are expressed in the nuclei of male mouse testes and female mouse reproductive organs [47]. The full-body TAp63 knockout mouse model appeared to retain normal ovary histology, including the follicles and oocytes [15]. However, oocytes from the TAp63-null or the p63-null mouse models were resistant to DNA damage typically induced by γ-irradiation or chemotherapy [15,21,29]. To investigate the functions of the p63 C-terminus, Lena et al. established a novel mouse model replacing p63α with p63β by deleting exon 13 of the *Trp63* gene to knock out the function of p63α [48]. 

Heterozygous female mice developed normal skin and thymus, two organs that canonically express p63α, but displayed ovarian dysfunction, atretic ovarian morphology, and follicle depletion. The ovarian phenotypes in these mice resembled those observed in humans with infertility. The oocyte loss observed in heterozygous ovaries was associated with the activity of TAp63β, which is rapidly degraded. The expression of TAp63β in this mouse model also triggered uncontrolled apoptotic death in primary oocytes independent of DNA-damage-driven phosphorylation. 

Previously reported p63 mutations related to human female fertility were also detected in this mouse model, lending credence to the hypotheses that the observed mechanism of fertility loss in mice could be similar to that in humans [48]. Additionally, by analyzing transactivation activity and the oligomeric state of other syndromic TAp63 mutants, Lena et al. predicted the oocyte fate for each TAp63α mutant. This study confirmed the importance of the p63 C-terminus in regulating murine oocyte integrity and the relationship between TAp63 and fertility.

### 6.2. Human

Human *TP63* mutations have been reported and classified into seven different diseases in the clinical setting, including five developmental disorders and two non-syndromic disorders. Point mutations in the DBD result in EEC (ectrodactyly-ectodermal dysplasia-cleft) syndrome, which is characterized by limb deformation, cleft lip and palate, and ectodermal dysplasia [49]. Mutations in the SAM and TID domains of the *TP63* C-terminus are responsible for ankyloblepharon, ectodermal defects, and cleft lip/palate syndrome (AEC) [50]. Patients with AEC suffer from severe skin erosions at or after birth. 

Limb-mammary syndrome (LMS), acro-dermato-ungual-lacrimal-tooth syndrome (ADULT) and EEC can be summarized as ELA due to the overlap in patient presentation, including ectodermal dysplasia, limb defects, and cleft lip and palate. ELA mutations are mainly located in the DBD, implying the inhibition of DNA binding and transactivation in the C-terminus [51]. Rapp Hodgkin syndrome (RHS), which displays overlapping clinical features with AEC, such as the presence of severe skin erosions, is often associated with mutations located in the SAM domain and TID of the C-terminus, which induces aggregation and, thus, inactivation of p63 [52]. 

The two non-syndromic disorders include isolated split hand/foot malformation type 4 (SHFM4), which only affects limb development, and non-syndromic orofacial cleft (nsOFC), which causes cleft lip. The mutations associated with these non-syndromic disorders are usually found throughout the sequence of *TP63* [53]. *TP63* mutations in human patients reported in the literature are believed to not only affect epidermal development but also impact female fertility (Figure 4).

#### 6.2.1. *TP63* Mutants in Syndromic Infertility

A 14-year-old female patient was described to have presented with all clinical LMS phenotypes as well as dysgenesis of the internal genitalia leading to infertility. Molecular analysis detected a 2-base pair (bp) deletion (c.1576-1577DelTT) in exon 13 of the *TP63* gene [54]. This frameshift mutation inserts a premature stop codon that modifies the α-isoforms of p63 without affecting other proteins. An abdominal sonogram and magnetic resonance imaging (MRI) showed a normal urinary tract with the absence of the uterus and ovaries. Hormonal profiling reflected hypergonadotropic hypogonadism and low plasma estrogen levels. 

Another family study reported three related female patients who presented with typical features of RHS/AEC and developed premature menopause around the age of 30 [55]. A heterozygous single nucleotide deletion (c.1783delC) was detected at codon 595 within exon 14 of the *TP63* gene. This mutation induces a modification of the reading frame and introduces a stop codon at base 1992 (TAA), 65 bp downstream to the canonic TGA stop. No other causes of premature menopause were identified in these patients. This reported mutation is located in the TID of the C-terminal region of p63, affecting the two α-isoforms of p63 (TAp63α and ∆Np63α) instead of the β- and γ-isoforms. 

Another case report described two adolescent sisters from an LMS family who presented with undetectable ovaries and hypoplastic uterine tissue, mammary glands, and nipples, which led to infertility [56]. In contrast to classically diagnosed LMS patients with limb defects, these two sisters developed normal hands and feet. With the use of exome sequencing, a novel nonsense variant (c.1927C>T, p.(Arg643*)) in exon 14 was identified and classified as pathogenic based on the pathogenicity evidence. The *TP63*: c.1927C>T, p.(Arg643*) mutation is located within the TID generating a truncated protein.

#### 6.2.2. *TP63* Mutants in Isolated POI

In addition to the above syndromic infertility cases, recent clinical studies have also reported a direct linkage between *TP63* mutations and isolated POI. By analyzing 13 POI patients, the genetic cause of POI was identified in one patient who presented with primary amenorrhea and was diagnosed with isolated POI. The variant (c.1780C>T p.(Arg594Ter)) was identified to be a de novo nonsense variant in *TP63* within the C-terminal exon and introduces a premature stop codon. The c.1780C>T p.(Arg594Ter) variant is located in the SAM domain and truncates *TP63* before the TID. 

The location of the c.1780C>T p.(Arg594Ter) variant results in nonsense-mediated decay in transcription and a truncated p63 protein. An additional 107 POI patients were recruited to further test the C-terminus of *TP63* for variants. A distinct *TP63* C-terminal truncating variant (c.1794G>A p.(Trp598Ter)) was detected in an unrelated patent who was diagnosed with isolated POI and primary amenorrhea with no additional causative variants. Other *TP63* variant-related syndromic malformations were not detected in these two POI patients [57].

One recent clinical study with 67 early POI patients described 37 ovary-related copy number variants (CNV) involving 44 genes. A 161 kilobase paternal intragenic duplication at 3q28 was identified in exons 2–10 of *TP63,* which produced an aberrant transcript without disrupting the reading frame. This intragenic duplication was located in the DBD, resulting in steady aberrant *TP63* mRNA in patients and, consequently, reduced p63 activity [58].

#### 6.2.3. Polymorphisms of *TP63*

Mutations in *TP63* have also been shown to decrease fertility for in vitro fertilization (IVF) patients [59]. Selected alleles were enriched in a single nucleotide polymorphism (rs17506395) of the *TP63* gene in IVF patients with advanced maternal age who were characterized to possess declined oocyte quality. This further confirmed the importance of p63 in maintaining oocyte quality and ovarian function.

## 7. Conclusions and Future Directions

The current clinical literature reporting *TP63*-related ovarian insufficiency is diverse in patient presentations and includes isolated POI and syndromic POI with RHS, LMS, and AEC. Except for the intragenic duplication, all *TP63* variants that have been associated with POI are located in the C-terminal region, which further confirms the involvement and importance of the *TP63* C-terminus in regulating ovarian development and function. Thus, the relative contribution of the C-terminus in TAp63 to female fertility should be thoroughly considered. 

The wide expression pattern and high expression levels of p63 typically observed in female reproductive organs and the clinical manifestations associated with the mutations of *TP63* in patients appear to indicate a scientifically relevant relationship. On the other hand, there are potential undiscovered SAM domain interactors or post-translational modifications abolished in ΔNp63 [60], allowing for the notion that epithelial tissue without ΔNp63 could be defective in the context of certain pathological conditions. Further studies of the morphogenesis of female reproductive organs will be required to fully explain the roles of p63 as well. 

Fertility preservation for cancer survivors is also an unmet need. Ovarian insufficiency and infertility are major side effects of conventional chemotherapy and radiation therapy. TAp63 expressed in oocytes can be activated by traditional cytotoxic cancer therapies, leading to uncontrolled oocyte apoptosis and follicle pool depletion. Future research may elucidate the molecular mechanisms of germ cell death to repair lesions incurred during cancer therapy. This will contribute to generating and developing effective fertoprotective agents to preserve female fertility. While the expression of p63 has been reported in male germ cells, the contribution of p63 in spermatogenesis remains unclear. Further research on p63’s function and capacity in reproduction will boost our understanding of the molecular mechanisms of female/male infertility in addition to oncogenesis.

## Figures and Tables

**Figure 1 ijms-22-08968-f001:**
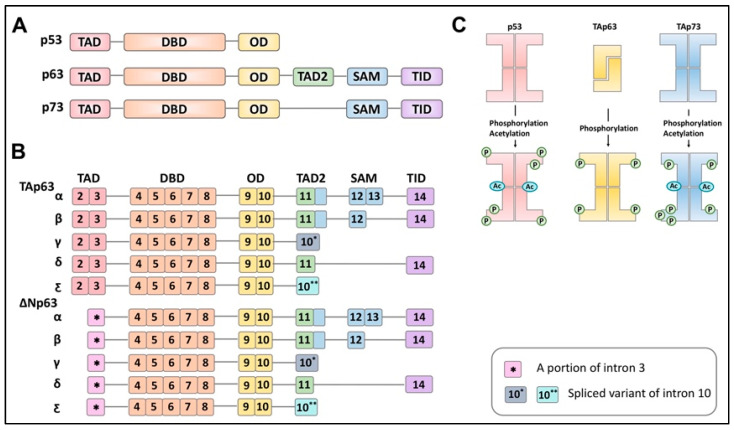
Schematic illustration of the p53, p63, and p73 protein structures and p63 isoforms. (**A**). p53 family protein structures. (**B**). Protein structures of p63 variants. (**C**). Post-translational modification of p53 family proteins. Transactivation domain (TAD), DNA binding domain (DBD), oligomerization domain (OD), sterile alpha motif domain (SAM), and transactivation inhibitory domain (TID).

**Figure 2 ijms-22-08968-f002:**
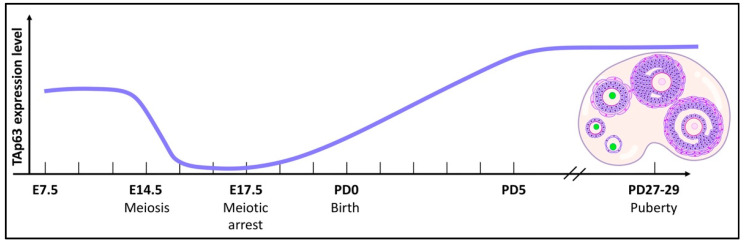
The TAp63 expression pattern. The graphical plot of TAp63 expression in primordial germ cells during embryonic development and in oocytes during folliculogenesis in the mouse. The green color indicates the expression of TAp63 in the oocyte nucleus. This is recapitulated based on Kurita et al., Suh et al., and Jurita et al.

**Figure 3 ijms-22-08968-f003:**
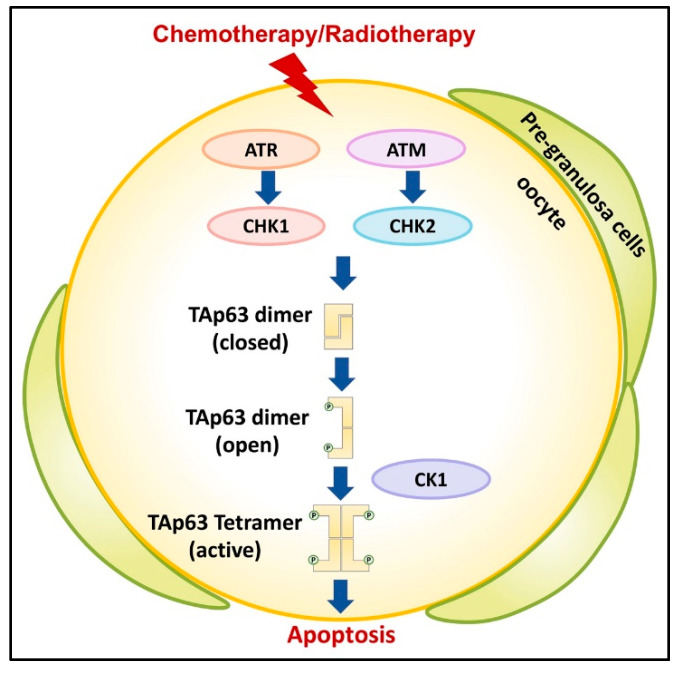
TAp63 is integral for DNA damage signaling in oocytes. Chemotherapy or radiotherapy activates the ATM/CHK2 or the ATR/CHK1 pathway. The phosphorylation of TAp63 by priming kinase CHK1/2 recruits a second executioner kinase CK1, which adds additional phosphorylation. This allows the opening of the closed dimeric state to eventually form an active tetrameric conformation. TAp63 acts as a transcriptional activator for downstream apoptotic signaling.

**Figure 4 ijms-22-08968-f004:**
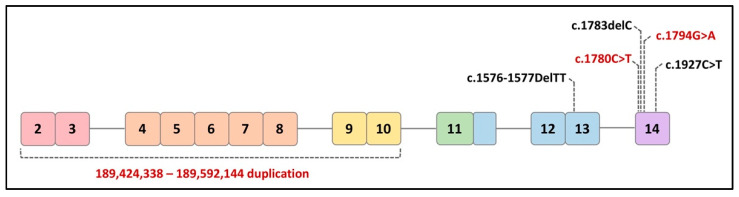
Distribution of mutations in *TP63α* associated with infertility in human patients. The red text indicates the isolated POI, and the black text indicates syndromic POI.

## Data Availability

Not applicable.

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
