# Peer review of "The Role of Mutant p63 in Female Fertility"

_ijms, 2021, doi:10.3390/ijms22168968_

Round 1

Reviewer 1 Report

Overall, the review is well written and illustrated. However, one purpose of the reviews is to give references to presented facts. And, this is not done well enough here. After the sentences that describe findings please put in references. Overall, the subject is interesting and will be well received.

Line 36 – Please incorporate this publication: p73 Is Required for Ovarian Follicle Development and Regulates a Gene Network Involved in Cell-to-Cell Adhesion. They also performed co-staining of p63 and p73

Line 68 – need reference the full TAp63a expression

Line 81 –  require the N-terminal TA domain.  Please put reference in.

108 -form-positive while… isoform?

Lines 128 -140 Again, a quite long paragraph with a single ref to (19). Is it all in this paper? I would check it up and add more refs.

Thus, meiotic DNA DSBs and 128 TAp63α expression appear at some level to be inversely correlated in newborn primary 129 oocytes [19]. Since predominantly all PD5 primary oocytes expressed only TAp63α and 130 lack γ-H2AX foci, either all meiotic DNA DSBs underwent repair by PD5 dictyate arrest 131 or those which failed to undergo repair underwent TAp63α-mediated death. Whereas 132 PD5 oocytes of the immature primordial follicles are totally obliterated within only 2 days 133 of ionizing radiation (IR) treatment, a significant number of PD0.5 immature oogonia pos-134 sess the capacity to survive IR-induced DNA damage for at least 7 days after IR treatment 135 due to the absence of TAp63. While irradiated PD5 oocytes demonstrate a prominent 136 TAp63 phosphorylation shift, irradiated PD0.5 oocytes did not. However, IR-induced γ-137 H2AX levels increased following IR treatment in both PD0.5 and PD5 ovary lysates, indi-138 cating that DNA damage was incurred in both PD0.5 and PD5 ovaries. Thus, oogonia 139 appear to quickly acquire the potential to induce TAp63α phosphorylation after birth.

Lines 141 – 147 Again, A single ref.

Line 251 and ovary. Ref

Line 277 Previously reported p63 mutations related 274 to human female fertility were also detected in this mouse model, lending credence to the 275 hypotheses that the observed mechanism of fertility loss in mice could be similar to that 276 in humans. (REF!!!!)

Line 281 – A paragraph 62.  without references at all.

Line 362 – it is not clear how this casual relationship was established.

Author Response

Reviewer 1

Overall, the review is well written and illustrated. However, one purpose of the reviews is to give references to presented facts. And, this is not done well enough here. After the sentences that describe findings please put in references. Overall, the subject is interesting and will be well received.

Thank you for the comment. The references were added accordingly to the review.

Line 36 – Please incorporate this publication: p73 Is Required for Ovarian Follicle Development and Regulates a Gene Network Involved in Cell-to-Cell Adhesion. They also performed co-staining of p63 and p73

Thank you for the suggestion. The suggested reference was added.

Line 68 – need reference the full TAp63a expression

Thank you for pointing this out. We added the reference in line 68.

Line 81 –  require the N-terminal TA domain.  Please put reference in.

Thank you for your remark. The corresponding reference was added.

108 -form-positive while… isoform?

Thank you for your comment. We added the references accordingly.

Lines 128 -140 Again, a quite long paragraph with a single ref to (19). Is it all in this paper? I would check it up and add more refs.

Thus, meiotic DNA DSBs and 128 TAp63α expression appear at some level to be inversely correlated in newborn primary 129 oocytes [19]. Since predominantly all PD5 primary oocytes expressed only TAp63α and 130 lack γ-H2AX foci, either all meiotic DNA DSBs underwent repair by PD5 dictyate arrest 131 or those which failed to undergo repair underwent TAp63α-mediated death. Whereas 132 PD5 oocytes of the immature primordial follicles are totally obliterated within only 2 days 133 of ionizing radiation (IR) treatment, a significant number of PD0.5 immature oogonia pos-134 sess the capacity to survive IR-induced DNA damage for at least 7 days after IR treatment 135 due to the absence of TAp63. While irradiated PD5 oocytes demonstrate a prominent 136 TAp63 phosphorylation shift, irradiated PD0.5 oocytes did not. However, IR-induced γ-137 H2AX levels increased following IR treatment in both PD0.5 and PD5 ovary lysates, indi-138 cating that DNA damage was incurred in both PD0.5 and PD5 ovaries. Thus, oogonia 139 appear to quickly acquire the potential to induce TAp63α phosphorylation after birth.

Thank you very much for pointing this out. We put the references accordingly as below.

Thus, meiotic DNA DSBs and TAp63α expressions appear at some level to be inversely correlated in newborn primary oocytes [21]. Since predominantly all PD5 primary oocytes expressed only TAp63α and lack γ-H2AX foci, either all meiotic DNA DSBs underwent repair by PD5 dictyate arrest or those which failed to undergo repair underwent TAp63α-mediated death [22]. Whereas PD5 oocytes of the immature primordial follicles are totally obliterated within only 2 days of ionizing radiation (IR) treatment, a significant number of PD0.5 immature oogonia possess the capacity to survive IR-induced DNA damage for at least 7 days after IR treatment due to the absence of TAp63 [5]. While irradiated PD5 oocytes demonstrate a prominent TAp63 phosphorylation shift, irradiated PD0.5 oocytes did not. However, IR-induced γ-H2AX levels increased following IR treatment in both PD0.5 and PD5 ovary lysates, indicating that DNA damage was incurred in both PD0.5 and PD5 ovaries [21]. Thus, oogonia appear to quickly acquire the potential to induce TAp63α phosphorylation after birth [16].

  1. Kim, S. Y.; Nair, D. M.; Romero, M.; Serna, V. A.; Koleske, A. J.; Woodruff, T. K.; Kurita, T., Transient inhibition of p53 homologs protects ovarian function from two distinct apoptotic pathways triggered by anticancer therapies. Cell Death Differ 2019, 26, (3), 502-515.
  2. Suh, E. K.; Yang, A.; Kettenbach, A.; Bamberger, C.; Michaelis, A. H.; Zhu, Z.; Elvin, J. A.; Bronson, R. T.; Crum, C. P.; McKeon, F., p63 protects the female germ line during meiotic arrest. Nature 2006, 444, (7119), 624-8.
  3. Kim, D. A.; Suh, E. K., Defying DNA double-strand break-induced death during prophase I meiosis by temporal TAp63α phosphorylation regulation in developing mouse oocytes. Mol Cell Biol 2014, 34, (8), 1460-73.
  4. Bolcun-Filas, E.; Rinaldi, V. D.; White, M. E.; Schimenti, J. C., Reversal of female infertility by Chk2 ablation reveals the oocyte DNA damage checkpoint pathway. Science 2014, 343, (6170), 533-6.

Lines 141 – 147 Again, A single ref.

Thank you for your critique. We added the references.

It has been demonstrated that oocytes in primordial follicles of mice lacking exons 2 and 3 of the Trp63 gene encoding the transactivation domain-specific N-terminus were resistant to equivalent doses of IR treatment that killed virtually all oocytes of primordial follicles in wild-type ovaries [12]. Similar results were observed when comparing wild-type ovaries grown and irradiated in vitro with ovaries isolated from TAp63-null mice. The association observed between TAp63α phosphorylation and oocyte cell death was the first to suggest that TAp63α was phosphorylated in response to DNA damage [16].

  1. Coutandin, D.; Osterburg, C.; Srivastav, R. K.; Sumyk, M.; Kehrloesser, S.; Gebel, J.; Tuppi, M.; Hannewald, J.; Schäfer, B.; Salah, E.; Mathea, S.; Müller-Kuller, U.; Doutch, J.; Grez, M.; Knapp, S.; Dötsch, V., Quality control in oocytes by p63 is based on a spring-loaded activation mechanism on the molecular and cellular level. Elife 2016, 5.

Line 251 and ovary. Ref

Thank you for your suggestions. The references were inserted as below.

Studies with mouse models revealed expression of TAp63 isoforms in all female reproductive organs, with the highest concentrations in the vagina and ovary [4, 17].

  1. Kurita, T.; Cunha, G. R.; Robboy, S. J.; Mills, A. A.; Medina, R. T., Differential expression of p63 isoforms in female reproductive organs. Mech Dev 2005, 122, (9), 1043-55.
  2. Nakamuta, N.; Kobayashi, S., Expression of p63 in the mouse primordial germ cells. J Vet Med Sci 2004, 66, (11), 1365-70.

Line 277 Previously reported p63 mutations related 274 to human female fertility were also detected in this mouse model, lending credence to the 275 hypotheses that the observed mechanism of fertility loss in mice could be similar to that 276 in humans. (REF!!!!)

Thank you for your critique. We added the references.

Previously reported p63 mutations related to human female fertility were also detected in this mouse model, lending credence to the hypotheses that the observed mechanism of fertility loss in mice could be similar to that in humans [50].

  1. Lena, A. M.; Rossi, V.; Osterburg, S.; Smirnov, A.; Osterburg, C.; Tuppi, M.; Cappello, A.; Amelio, I.; Dotsch, V.; De Felici, M.; Klinger, F. G.; Annicchiarico-Petruzzelli, M.; Valensise, H.; Melino, G.; Candi, E., The p63 C-terminus is essential for murine oocyte integrity. Nat Commun 2021, 12, (1), 383.

Line 281 – A paragraph 62.  without references at all.

Thank you so much for your critique. The references were added to this paragraph.

Human TP63 mutations have been reported and classified into 7 different diseases in the clinical setting, including 5 developmental disorders and 2 non-syndromic disorders. Point mutations in the DBD result in EEC (ectrodactyly-ectodermal dysplasia-cleft) syndrome, characterized by limb deformation, cleft lip and palate, and ectodermal dysplasia [51]. Mutations in the SAM and TID domains of the TP63 C-terminus are responsible for ankyloblepharon, ectodermal defects, and cleft lip/palate syndrome (AEC) [52]. Patients with AEC suffer from severe skin erosions at or after birth. Limb-mammary syndrome (LMS), acro-dermato-ungual-lacrimal-tooth syndrome (ADULT) and EEC can be summarized as ELA due to overlap in patient presentation, including ectodermal dysplasia, limb defects, and cleft lip and palate. ELA mutations are mainly located in the DBD, implying inhibition of DNA binding and transactivation in the C-terminus [53]. Rapp Hodgkin syndrome (RHS), which displays overlapping clinical features with AEC, such as the presence of severe skin erosions, is often associated with mutations located in the SAM domain and TID of the C-terminus which induces aggregation and thus inactivation of p63 [54]. The two non-syndromic disorders include isolated split hand/foot malformation type 4 (SHFM4) which only affects limb development and non-syndromic orofacial cleft (nsOFC) which causes cleft lip. The mutations associated with these non-syndromic disorders are usually found throughout the sequence of TP63 [55]. TP63 mutations in human patients reported in the literature are believed to not only affect epidermal development but also impact female fertility (Fig. 4).

  1. Wawrzycki, B.; Pietrzak, A.; Chodorowska, G.; Filip, A. A.; Petit, V.; Rudnicka, L.; Dybiec, E.; Rakowska, A.; Sobczyńska-Tomaszewska, A.; Kanitakis, J., Ectrodactyly-ectodermal dysplasia-clefting syndrome with unusual cutaneous vitiligoid and psoriasiform lesions due to a novel single point TP63 gene mutation. Postepy Dermatol Alergol 2019, 36, (3), 358-364.
  2. Tajir, M.; Lyahyai, J.; Guaoua, S.; El Alloussi, M.; Sefiani, A., Ankyloblepharon-ectodermal Defects-cleft Lip-palate Syndrome Due to a Novel Missense Mutation in the SAM Domain of the TP63 Gene. Balkan J Med Genet 2020, 23, (1), 95-98.
  3. Prontera, P.; Garelli, E.; Isidori, I.; Mencarelli, A.; Carando, A.; Silengo, M. C.; Donti, E., Cleft palate and ADULT phenotype in a patient with a novel TP63 mutation suggests lumping of EEC/LM/ADULT syndromes into a unique entity: ELA syndrome. Am J Med Genet A 2011, 155a, (11), 2746-9.
  4. Bougeard, G.; Hadj-Rabia, S.; Faivre, L.; Sarafan-Vasseur, N.; Frébourg, T., The Rapp–Hodgkin syndrome results from mutations of the TP63 gene. European Journal of Human Genetics 2003, 11, (9), 700-704.
  5. Clements, S. E.; Techanukul, T.; Holden, S. T.; Mellerio, J. E.; Dorkins, H.; Escande, F.; McGrath, J. A., Rapp-Hodgkin and Hay-Wells ectodermal dysplasia syndromes represent a variable spectrum of the same genetic disorder. Br J Dermatol 2010, 163, (3), 624-9.

Line 362 – it is not clear how this casual relationship was established.

Thank you for your suggestion. The sentence was revised as:

The wide expression pattern and high expression levels of p63 typically observed in female reproductive organs and the clinical manifestations associated with the mutations of TP63 in patients appear to indicate a scientifically relevant relationship.

Reviewer 2 Report

Abstract: Well written, no comment.

Introduction: Well written, no comment.

Structure of the p53 family members: No comment.

p63 expression and regulation in the ovarian germ cells:

 If you can rewrite the sentence, line 92-94.

Line 96, can you explain what is E10.5?

Line 127, P0.5 and in line 138, PD0.5, are they same?

Regulation of p63 structure in the oocyte: No comment.

Regulation of p63 in the oocyte:

In line 184, and 185, are they mean gene and protein?

In line 201, Kim Dal-Ah et al, year of publication not mentioned.

P63 and female infertility:

Line 284, full form for DBD?

In human, not many cases so far observed where p63 is involved in female fertility. But definitely more work need to be done. Overall good work.

Conclusions and future directions: no comment.

Author Response

Reviewer 2

Abstract: Well written, no comment.

Introduction: Well written, no comment.

Structure of the p53 family members: No comment.

p63 expression and regulation in the ovarian germ cells:

 If you can rewrite the sentence, line 92-94.

Thank you for your critique. The sentence was edited to avoid any confusion.

Our current understanding of the biological roles of TAp63, commonly referred to as the guardian of the female germline, in the genomic integrity of female germ cells has been mostly obtained using mouse models.

Line 96, can you explain what is E10.5?

Thank you for your question. The E10.5 indicates embryonic day 10.5. The acronym of ‘E’ has been written when it first time appeared.

‘In mice, TAp63 expression has been detected in diploid mitotic primordial germ cells (PGCs) as early as embryonic day 7.5 (E7.5), after which PGCs migrate to the genital ridge at E10.5 via the hindgut.’

Line 127, P0.5 and in line 138, PD0.5, are they same?

Thank you so much for pointing this out. The P0.5 was revised to PD0.5.

Regulation of p63 structure in the oocyte: No comment.

Regulation of p63 in the oocyte:

In line 184, and 185, are they mean gene and protein?

Thank you for your questions. Puma and Noxa (italic) represent their gene names, while PUMA, NOXA, BAX, and BAK (capital) mean protein names.

In line 201, Kim Dal-Ah et al, year of publication not mentioned.

Thank you for the remark. The corresponding reference was listed right after the sentence and the year of publication is 2014. To avoid any further confusion, we revised this sentence.

P63 and female infertility:

Line 284, full form for DBD?

Thank you for your question. The DBD represents DNA-Binding Domain, and this information was provided in line 53 when it first time appeared.

In human, not many cases so far observed where p63 is involved in female fertility. But definitely, more work need to be done. Overall good work.

Conclusions and future directions: no comment.
